

# Integration between X-Band Radar and Buoy Sea State Monitoring

Giovanni Ludeno [1,*], Ferdinando Reale [2], Francesco Raffa [3], Fabio Dentale [2], Francesco Soldovieri[1], Eugenio Pugliese Carratelli [2] and Francesco Serafino [4]

[1]Institute for Electromagnetic Sensing of Environment (IREA), National Research Council (CNR); ludeno.g@irea.cnr.it; soldoviere.f@irea.cnr.it

[2]CUGRI (University Centre for Research on major Hazard) and MEDUS (Maritime Engineering Division University of Salerno); freale@unisa.it ; fdentale@unisa.it ; epc@unisa.it

[3]Institute for Coastal Marine Environment (IAMC), National Research Council (CNR); francesco.raffa@cnr.it

[4]CNR-IBMET (Institute of Biometeorology of National Research Council); serafino.f@irea.cnr.it

*Correspondence: ludeno.g@irea.cnr.it; Tel.: +39 081 7620654

**Abstract.** The paper presents the results of an integrated buoy and X-Band radar sea state monitoring activity carried out on the southern coast of Sicily. The work involved the integration of buoy and radar data, as well as the simultaneous acquisition of Significant Wave Height (SWH) values from two similar radar sets located at a slight distance from each other – a rare and fortunate circumstance which took place during two storms in the winter 2014-2015. Good consistency and repeatability was reached between the two radars and the reliability of X-Band radar as a wave monitoring system was confirmed by the comparison with the buoy wave meter. A further and important result of the work is the knowledge gained on the short spatial and temporal fluctuations of the sea state: while such Small Scale Storm Variations (SSSV) cannot be easily discriminated from electromagnetic effects and from algorithm artefacts, some important progress has been done towards the identification of this phenomenon. Integration of different sensors is the key to a definite improvement of sea state monitoring for most coastal applications.

## 1 Introduction

There can be no doubt about the ever growing importance of coastal monitoring from all points of view: biological, chemical, water quality, temperature, currents, etc. Among all the relevant parameters those concerning wave agitation ("the sea state") are, if not the most important, certainly the first to be considered, since they influence all the others, directly or indirectly. They also closely connected to the meteorological conditions, both on a global and a local scale.

Wave height measurement is probably the most important monitoring activity being carried out on the sea. It has a great scientific and economical relevance [1,2] therefore an equally great effort is dedicated all over the world to the maintenance of complex network of measuring systems. Routine sea wave monitoring, is presently carried out by accelerometric or GPS buoys deployed mostly along the coasts. Satellite altimeters [3] and SAR radars [4,5], also play a very important role in offshore



monitoring, but their role is constrained by the long intervals of their passages. In the last two decades, the X-Band marine radar systems have been employed as a remote sensing tool for the sea state monitoring both on ships and on land as an example harbours [6-12]. Each technique has its merits and its drawbacks and no single technique can claim a clear cut superiority to the others; integration between them, will probably be the final answer to worldwide wave agitation monitoring [1,2]. The

application and the improvement of radar technology for sea wave measurement is thus bound to be a hot topic in the next few years, even though a great amount of effort has of course already been dedicated to the its development. Given this, it is somewhat surprising that so few experiments are reported in the literature of combined application of two or more systems. Also, it must be remarked that most of the published results for land based wave radar, has been concentrated on experiments carried out in the Oceans or on the North Sea [10-15], while much less work has been carried in enclosed or semi-enclosed sea

such as the Mediterranean [8,9, 16-19]. The latter environment is much more challenging for wave measurement, since the combination of quickly varying wind fields and short and irregular fetches produces complex wave fields and often non typical spectra. For instance, the distinction between swell and wind wave is much less clear than in the oceans, and the periods and the wavelengths are generally much shorter than in large seas: a higher space resolution is therefore necessary, and testing and calibration require a special care. The present paper presents the work carried out in a particular location on the South coast of

Sicily, where for short period it has been possible to carry out experiments with the contemporaneous presence of two X-Band wave radar and a buoy wave meter.

The paper is organised as follows: a theoretical background is described in section 2; the basic theory and the computational algorithm are briefly described in section 3; the location and the experimental set-up are described in section 4; results are discussed in section 5 and the conclusion are drawn in section 6.

**2 Theoretical Background**

The most commonly used methods to derive Significant Wave Height (Hs) from radar images are based upon the analysis of the zero-order moment of the wave spectrum and on the analysis of the Signal to Noise Ratio (SNR) between the spectral wave energy and the background noise component [15,20-22]. Another strategy, which also makes use of the peak wavelength and the mean wave period, is based on multilayer perceptions [23]. However the problem of estimating the Hs from radar data is

still not completely solved since most of the methods require some data provided by external references to be used for calibration purposes. A way to overcome this issue is shown in [24], where the Authors proposed a method to get the Significant Wave Height from the analysis of shadowing occurring in radar images, without any reference measurement.

Remocean X-Band sea radar has been developed from the beginning in the Mediterranean environment, even though significant tests have been carried out also in the North Sea [13]. Wave measurements have also been carried at the Giglio

Island, in Italy, where the Remocean system has been employed to both monitor the sea conditions and to support the removal activities of the Costa Concordia wreckage, a cruise ship that capsized and partially sank close to the Giglio harbour in January 2012 [17]. While carrying out this work the Authors tested the effectiveness of the sea wave height reconstruction by



comparing the X-Band radar estimated sea state parameters and 2-D directional spectra with both buoy measurements and the results of the wave models WW3(Wave Watch III) and SWAN (Simulating Wave Nearshore). In addition, the 3-D (space-time) reconstruction of the sea surface showed the capability of the wave radar system to detect sea waves as reflected by the Costa Concordia shipwreck. This capability has been further investigated in [17], where Authors have again shown the possibility to identify reflected sea waves by analysing the radar spectra.

As stated above this paper describes a further advancement of the testing and validating procedure of the system, based on the integration and the comparison of results experiments gathered in shallow waters along the southern Coast of Sicily, during two different winter storms. In the first episode, offshore wave data were collected by a wavemeter buoy of the Italian State Network (RON), which allowed a proper comparison of the data. During the second storm no buoy data were available, but the test were carried out by comparing two radar installations, located a few miles apart; this provided a unique possibility of verifying the coherence and the repeatability of X-Band radar wave measurements.

Wave analysis data produced by ECMWF (European Centre for Medium range Weather Forecast) are also shown as a useful reference; such data, regularly produced on the basis of global and regional numerical weather and wave modelling and satellite data acquisition, provide a rough guidance to the actual sea state. The availability of wave radar installations such as those considered here will certainly help to improve the quality of such data.

## 3 Data Processing Approach

The basic theory is well known, and it can be summarized as follows:

according to the linear (Airy) wave theory, the sea elevation profile $\eta(\underline{r}, t)$ can be regarded as the superposition of monochromatic waves with different amplitudes A, periods T, directions $\hat{k}$ and wavelengths λ, i.e.

$$\eta(\underline{r}, t) = \iint_K \int_\Omega Z(\underline{k}, \omega) e^{j(\underline{k}\,\underline{r} - \omega t)} \, d\underline{k} d\omega, \tag{1}$$

where t is the observation time and $\underline{r} = (x, y)$ represents the position vector, while $\omega = 2\pi/T$ is the angular frequency, $\underline{k} = k\hat{k} = (k_x, k_y)$ is the wave vector $k = |\underline{k}| = 2\pi/\lambda$ begin the wave number, and $Z(\underline{k}, \omega) = A(\underline{k}, \omega) e^{j\phi(\underline{k}, \omega)}$ is the complex amplitude of each component [7, 20]. The angular frequency ω and the wave number $k$, (i.e. T and λ) are not arbitrary, as they are tied to each other by the physical properties of sea gravity waves (dispersion equation):

$$\omega(\underline{k}) = \sqrt{gk \tanh(kh)} + \underline{k}\,\underline{U}, \tag{2}$$

where $g$ represents the acceleration due to the gravity, $h$ is the (supposedly known) bathymetry an $\underline{U} = (U_x, U_y)$ dis the sea surface current. Useful synthetic information of the ocean wave state can be directly retrieved from the above mentioned spectral components by computing the spectral moments

$$m_n = \frac{1}{2\pi} \iint_K \int_\Omega \omega^n \langle |Z(\underline{k}, \omega)|^2 \rangle \, d\underline{k} d\omega, \tag{3}$$

where $\langle \cdot \rangle$ stands for the statistical expectation operator, and n is a non-negative integer.



A particularly important parameter is the Significant Wave Height Hs, related to the spectral moment of zero order by $H_s = 4\sqrt{m_0}$. Estimating this latter quantity is anything but a straightforward task, since it requires the retrieval of the sea spectrum $\langle |Z(\underline{k},\omega)|^2 \rangle$ from distorted radar measurements. The mapping of sea waves into radar data is indeed affected by several distortions, known as modulation effects, so that radar echoes do not represent the sea elevation profile, as they are

tied instead to the slope of the long waves (tilt modulation) and to the roughness of the riding ripples (hydrodynamic modulation).

The analysis of radar data provides useful information about the sea state, e.g. direction, period and wavelength of the dominant waves, as well as the surface current within the radar observation space. The opportunity to estimate the sea current fields derives from the propagation mechanisms of the sea gravity waves as ruled by the dispersion relation Eq. (2) reported above.

The data processing approach adopted in this work is the "Local Method" proposed in [13, 16-19], which reconstructs inhomogeneous surface current fields from X-Band radar data. The inversion procedure can be summarized as in the block diagram in Figure 1.

Each radar image belonging to the temporal sequence considered is partitioned into Ns spatially overlapping sub-areas, thus providing Ns temporal sub-sequences. After such data partitioning the Ns radar spectra, each relevant to a given sub-area, are

computed via the FFT algorithm [13, 17, 19].

Each spectrum $\{F^j(\underline{k},\omega)\}_{j=1,\dots,N_s}$ is then processed via the Normalized Scalar Product (NSP) method [8], as to retrieve the local surface current vector through the following estimator:

$$V^j(\underline{U}) = \frac{\langle |F^j(\underline{k},\omega)|, G(\underline{k},\omega,\underline{U}) \rangle}{\sqrt{P_F P_G}}, \tag{4}$$

where $G(\underline{k},\omega,\underline{U}) = \delta(\omega - \sqrt{gk \tanh(kh)} + \underline{k}\,\underline{U})$ is the characteristic based on the dispersion relation ($\delta$ is the Dirac delta distribution), $\langle |F|, G \rangle$ represents the scalar product between the functions $|F|$ and G, while $P_F$ and $P_G$ are the powers associated

to $|F|$ and G respectively. The knowledge of the sea surface currents field $\underline{U} = (U_x, U_y)$ is required to define the band-pass filter; if no a-priori information is available for these quantities, they have to be estimated directly from radar data [6, 8, 11, 13].

The Band-Pass (BP) filter is built on the basis of the dispersion relation defined in Eq. (2) and then applied to the image spectrum $F(k_x, k_y, \omega)$; the result of the filtering procedure is the function $\tilde{F}(k_x, k_y, \omega)$. An equalization step is implemented

using the spectral Modulation Transfer Function (MTF), is to move from the filtered radar image spectrum $\tilde{F}(k_x, k_y, \omega)$ to the desired sea-wave spectrum $F_w(k_x, k_y, \omega)$ [17,20] by minimizing the electromagnetic modulation effect. After the filtering and equalization steps, wave parameters (direction, period and wavelength of the dominant waves as well as the Significant Wave Height) can be retrieved from the analysis of the calculated sea directional spectrum [17, 19-21].



## 4 Location and Data

The work described in this paper was carried out by integrating the data collected by two X-Band wave radar stations and by an accelerometric wave buoy (Mazara, courtesy of ISPRA, Italian Environmental Agency).

The two wave radar system were installed on the Southern Coast of Sicily (Figure 2): Sciacca (Agrigento) and Cape Granitola

(Trapani), at about 33 km away from each other in line of sight. The Cape Granitola system is based on a CONSILIUM X-Band radar radiating a maximum power of 12.5 KW and equipped with a 9-ft (2.74 m) long antenna, while a SPERRY Marine X-Band radar with an 8-ft (2.4 m) long antenna and radiating a maximum power of 25 kW is located at the Sciacca site.

Both systems operate in the short pulse mode (i.e., pulse duration of about 50 ns) at 9.5 GHz (X-Band) and the pulses are transmitted and are received in the horizontal plane (HH polarization). Each wave radar system is connected to a radar interface,

which incorporates an analog-to-digital (AD) converter for the received signal. The radar images are stored using a 13-bit unsigned integer format, on a $1024 \times 1024$ pixels Cartesian grid. The details of the parameters of acquisition for both wave radar systems are reported in Table 1.

The data set available for the two events considered are the following:

*   Event I: (CAPO GRANITOLA 21-22 October 2014). Available data: half hourly Significant Wave Heights (Hs),

direction and peak period from the ISPRA Mazara buoy from 00:30 to 12:00 of the 22nd; hourly radar significant wave heights at the Granitola location, direction and peak period from the 22nd at 01:00 to the 23rd at 01:00; 6 hours ECMWF analyses on grid points of coordinates coordinates 37.5° N , 12.625° E  and   37.5° N, 12.75° E, indicated as P10 and P11 in Fig2.

*   Event II (5-6March 2015). Available data: Hourly radar Significant Wave Height, direction and peak period from

2015, March the 5th at 00:00 to March the 6th at 24:00, both at the Granitola and Sciacca locations;. 6 hours ECMWF analyses on P10 and P11.

## 5 Analysis

### 5.1. Event I (22nd October 2014 Cape Granitola)

Buoy and radar observations have been taken hourly at roughly the same times (there is occasionally a 10' difference

between the two series) and they overlap between 0:30 to 12:00 of the 22nd of October 2014, basically coinciding with the raising part of a storm. Figure 3 reports all useful data.

ECMWF (analysis) results, also reported in Figure 3, obviously underestimated the storm intensity. The gap between models and measurements is quite common and derives mostly from the uncertainty of the weather forecasting systems which drive the large scale wave models Some aspects of this problems are discussed in [25,26] and the

availability of reasonably cheap and efficient measurement systems such as X-Band radar provides an ideal tool to calibrate and verify modelling chains. Some radar values are missing and have therefore been eliminated together with the corresponding buoy data.





The value of the radar data is related to the electromagnetic backscatter of the sea surface rather than to the wave elevation itself. In particular, the wave spectrum retrieved from the analysis of the radar data represents a scaled version of the actual wave spectrum and the estimation of Hs requires a calibration step aimed at retrieving the quoted scale factor, which depends on the particular wave radar installation [15,20].

Since the Mazara wave meter buoy and the Cape Granitola radar test area are located at some distance (about 9 nm) and at a different depth from each other (about 100m for the buoy and about 25 m for the radar areas), a linear wave transformation of the buoy wave height data was carried out to provide a reliable reference.

After carrying out these transformations, an important result was obtained by correlating radar and buoy Hs values (Figure 4).

As it can be seen the correlation is very high, and the radar provides an excellent estimate of the buoy Hs, at least for Hs greater than 1m. It is also easy to remark that a linear correlation only holds for Hs greater than about 1m; this does not represent a big drawback anyway. Further important information can be gathered from the scatter of data points along the regression line; as it has been shown elsewhere, this is an indicator of short term storm intensity variations [26]. Figure 5 shows the available results.

The standard deviation MSE of Hs both radar and buoy is given by Equation 5:

$$MSE = \sqrt{\frac{\sum_{i=1}^{N}\left(H_{s_i}-L_i\right)^2}{(N-1)}},\tag{5}$$

where the $L_i$ are the values of the linear regression at the same time, and N the number of available measurements. A scatter index CV = MSE/M is also considered, M being the mean of all the N values of the available data (13 in the case considered here). It is immediately evident that the scatter of the radar measured values is very similar to that of
20 the buoy values. Table 2 summarizes the results so far.

This scatter (CV) is certainly in part due to measurement errors, but it also provides some evidence of the presence of random oscillations of the storm intensity (Small Scale Storm Variations, SSSV). Such SSSV, which are too small in space and time scale to be forecasted or analysed by the wave generation models driven by Numerical Weather Prediction System, have been shown and measured by analysing Satellite altimeter SAR and buoy data [25], but so
far an important issue is the difficulty of differentiating between the effective oscillation and the inevitable instrumental errors. The similar values for MSE and CV of the very same storm with two entirely different measuring systems opens new possibilities for the analysis of SSSV. Unfortunately, very little can be said about the remaining part of the storm, since the wave meter buoy was not operating at the time; the only useful information are the MSE, M and CV values, as above which are shown in Figure 6, and appear to be consistent with the previous ones.

*5.2. Event II (Cape Granitola and Sciacca 5-6 March 2015)*



As stated above, no buoy data were available for this episode; however the simultaneously availability of two separate sets of radar measurement at a few miles' distance offers a rare opportunity to verify the consistency and the repeatability of X-Band radar wave measurement. Figure 7 reports the Hs measured during the main storm recorded during that period, as well as the ECMWF analysis – which appears here to be consistent with the radar data.

Here again some data are missing, in particular around the peak of the event; elaborations similar to those shown above can be carried out by considering the correlation between the two radar system rather than the correlation with a buoy wave meter. Figure 8 provides the results for two subsets of data, i.e. the ascending and the descending part of the storm.

Some improvement can be obtained by considering that the two radar areas are located at a considerable distance
from each other: since the prevailing direction of the storm is from about 250°, the wave trains reach the area off-shore of Cape Granitola (upwind) about 22 NM (40 km) before they reach Sciacca. Assuming an average depth of 25 m, the group velocity of the waves would be of about 16 m/s, thus implying that the agitation off Sciacca would be about 50 minutes late compared to off-shore Cape Granitola; it does therefore make sense to compare the data at the two locations by shifting the Sciacca data by about one hour (see Figure 9).

The correlations between the two set of data after such a shift show a remarkable improvement, specially so in the descending branch (Figure 10).
Once the reliability of the results is confirmed, it is worth considering again the oscillation around the trends (see Figure 11).

Table 3 summarizes the results so far.

The MSE and the CV values of Table 3 of the radar data should be compared with the similar results in Table 2. Since some amount of small scale oscillation around a trend is a normal feature of real sea states, the inherent and instrumental error of X-Band radar can be shown to be quite small, even though it is impossible to estimate its effective value.

## 6 Conclusions

While X-Band radar measurement of the sea state is a well established techniques in the oceans and in the open seas, its application in enclosed seas and in near shore conditions is far from being a settled problem. The work described in this paper has involved the integration of buoy and radar data, as well the simultaneous acquisition from two similar radar sets. New elements of confidence have therefore been gained on the reliability of X-Band radar hourly measurement of the Significant Wave Height even in critical conditions; besides, a good degree of consistency and repeatability was reached between two
radar data sets at a slight distance from each other.

A further useful result is linked to fluctuations of the sea state: they are certainly an index of the measurement error, but they also include an important component due to real and effective oscillation of the significant wave height, which is far from constant. Small Scale Storm Variation (SSSV) cannot be easily discriminated from E/M effects and from algorithm artefacts,



but the consistency of such oscillation as revealed by both buoys and radar is an important step forward in identifying this phenomenon. Integration of all the available sensor is therefore the key to a definite improvement of sea state data for most coastal application.

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



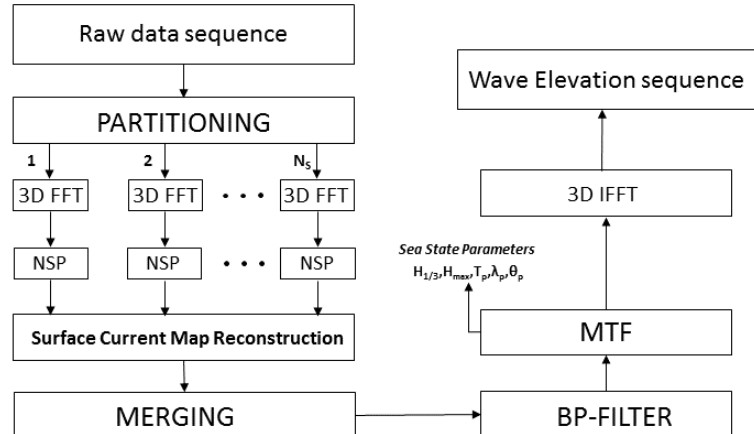

**Figure 1.** Block diagram of inversion procedure.

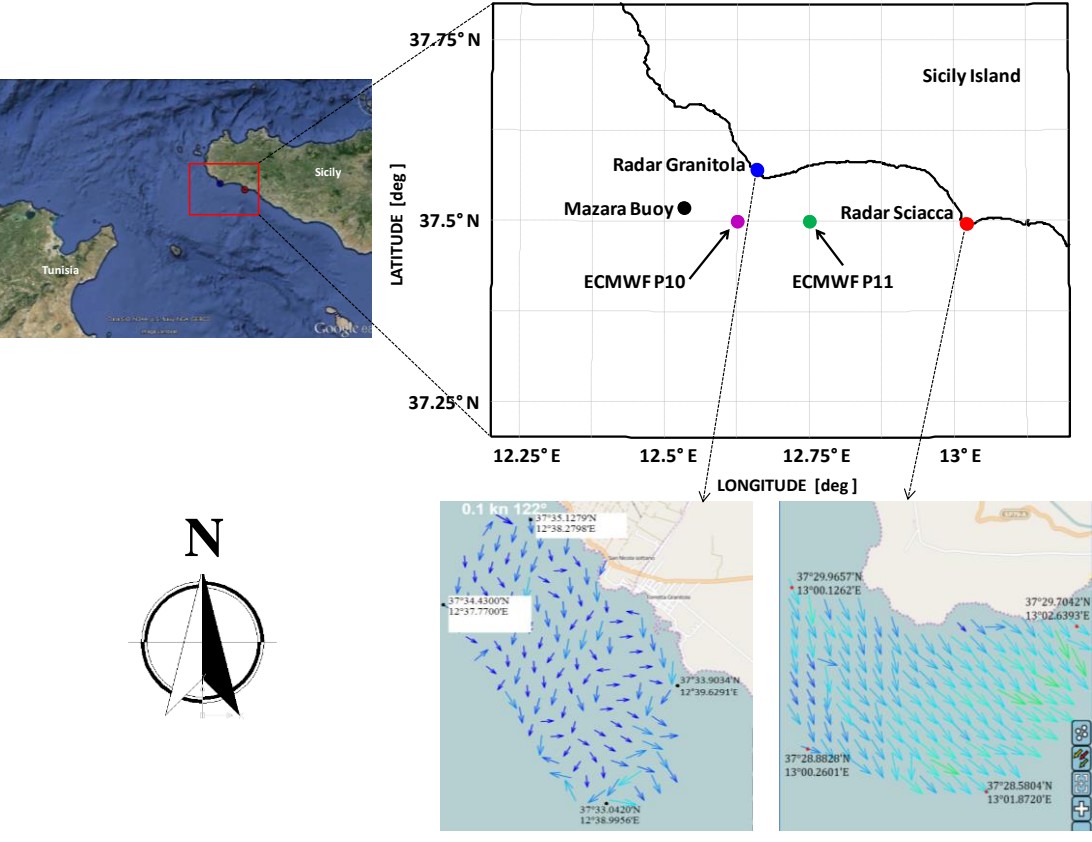

**Figure 2.** Test site location. Sciacca and Granitola radar stations are shown together with the Mazara wave meter buoy (RON) and two ECMWF grid points. The radar coverage is also shown in the lower inserts.




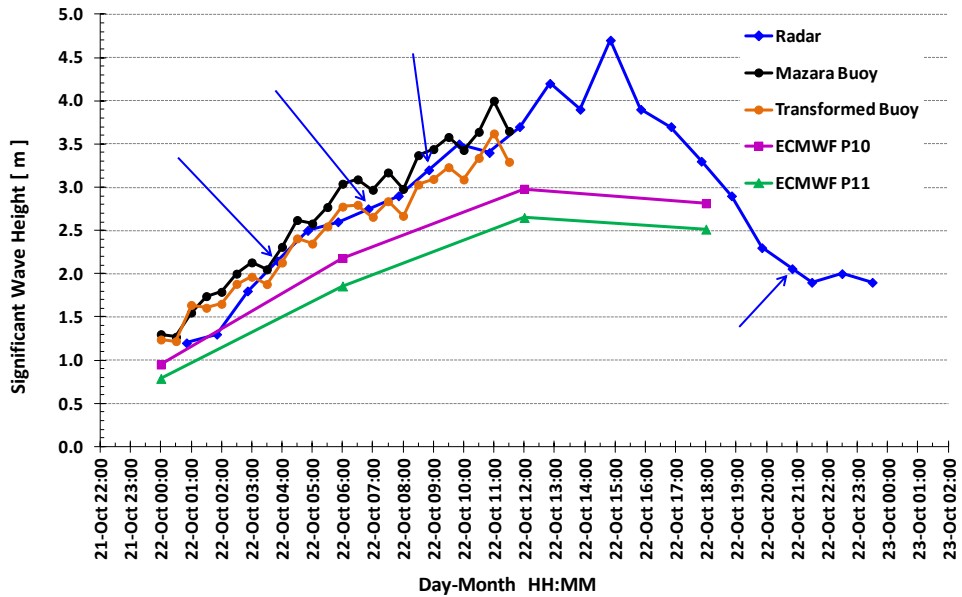

**Figure 3.** Significant Wave Height values from all sources. Buoy data are reported both as originally provided by RON and in their transformed values at a depth of 25 m. Missing radar data are interpolated with their neighboring values and indicated by the arrows.

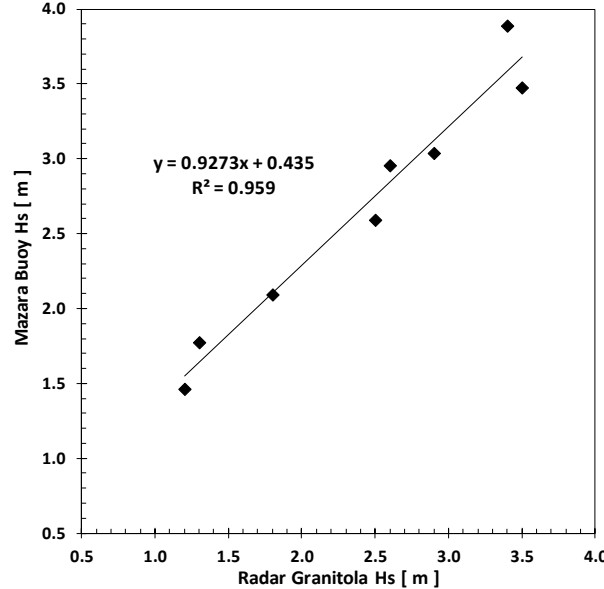

**Figure 4.** Correlation between buoy and radar Significant Wave Height values





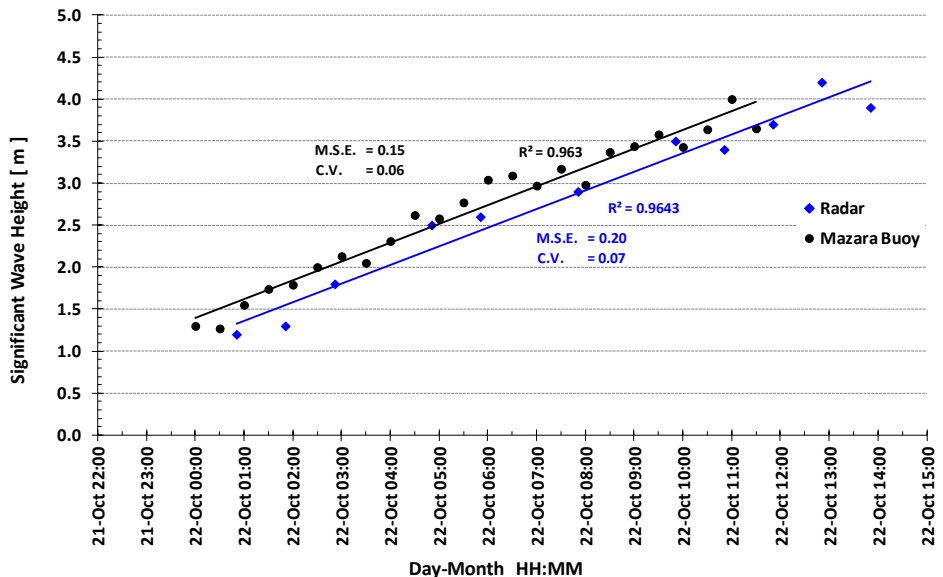

**Figure 5.** Best fit and scatter of all data during storm growth

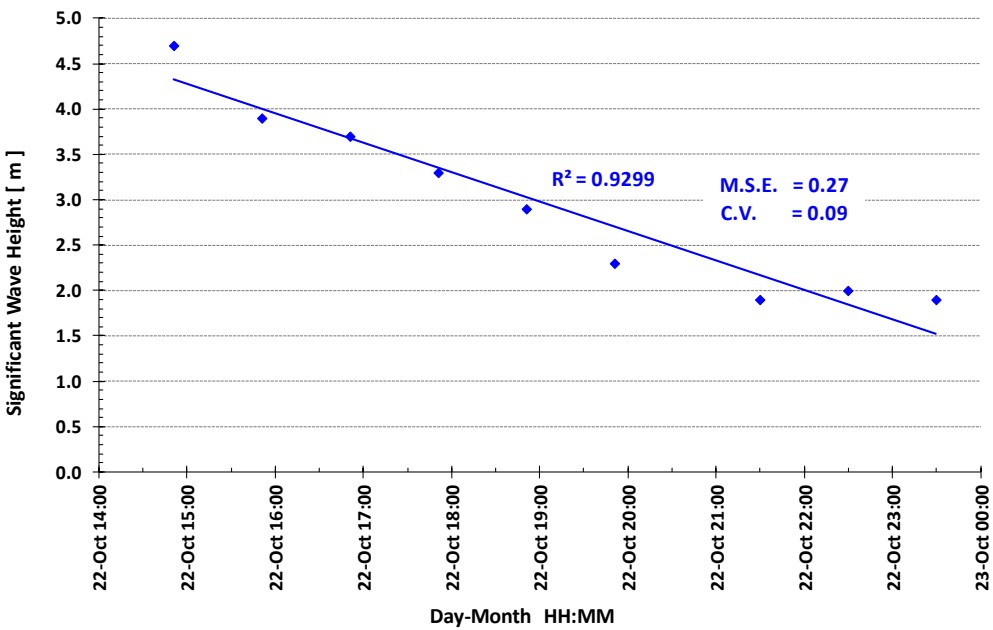

**Figure 6.** Best fit and scatter of radar data during storm decline.




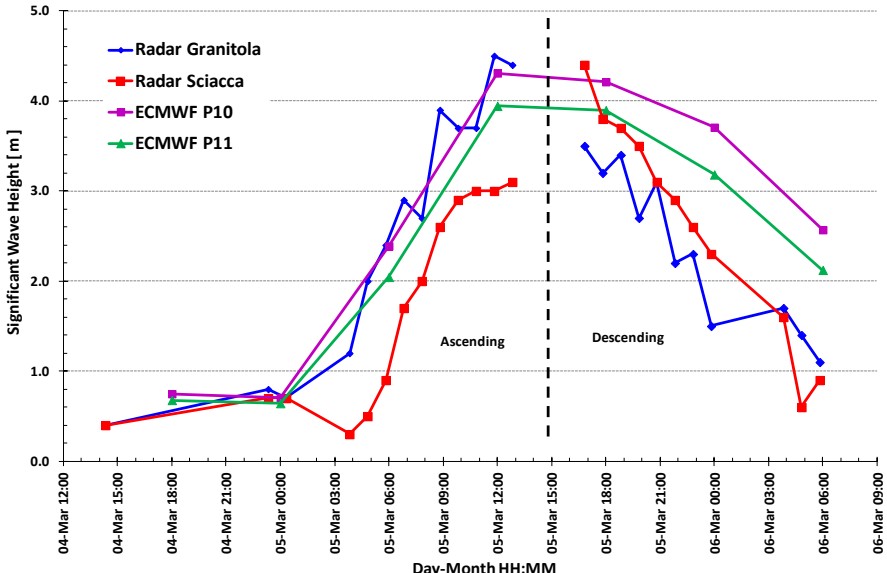

**Figure 7.** Significant Wave Height values from Sciacca and Granitola radar measurements. Vertical dashed line divides the storm growth (left) from the decline (right).

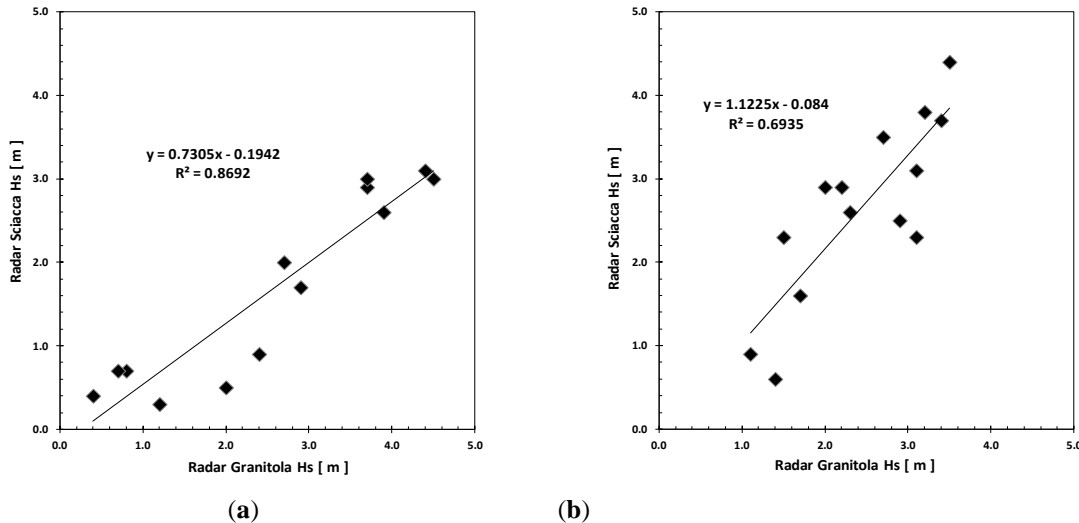

**Figure 8.** Correlation between Cape Granitola and Sciacca radar measurements: (**a**) storm growth; (**b**) storm decline.



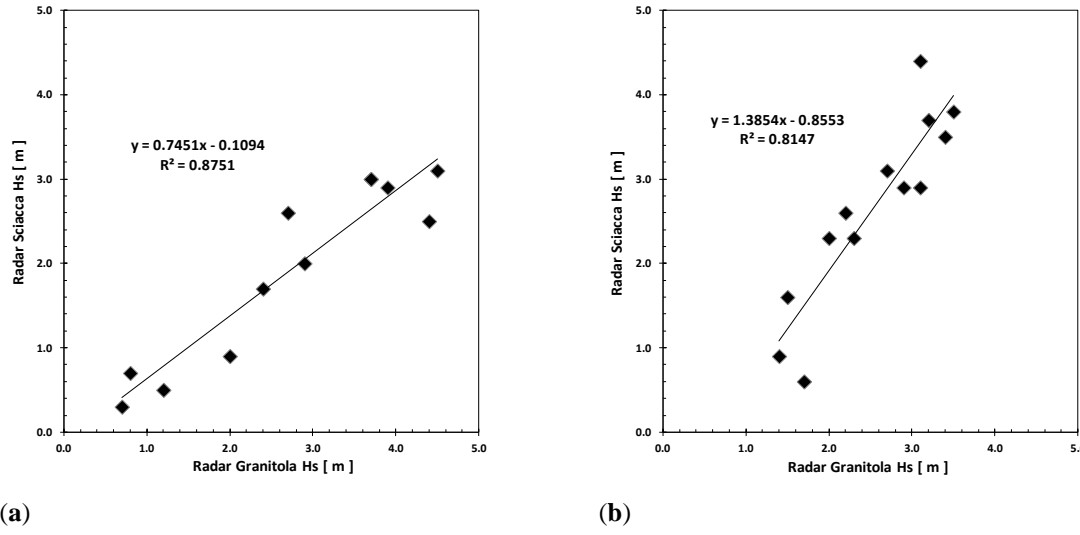

**Figure 9.** Cape Granitola (shifted 1 hour forward) and Sciacca radar data.

**Figure 10.** Correlation between Cape Granitola (shifted 1 hour forward) and Sciacca radar measurements: (**a**) storm growth; (**b**) storm decline.



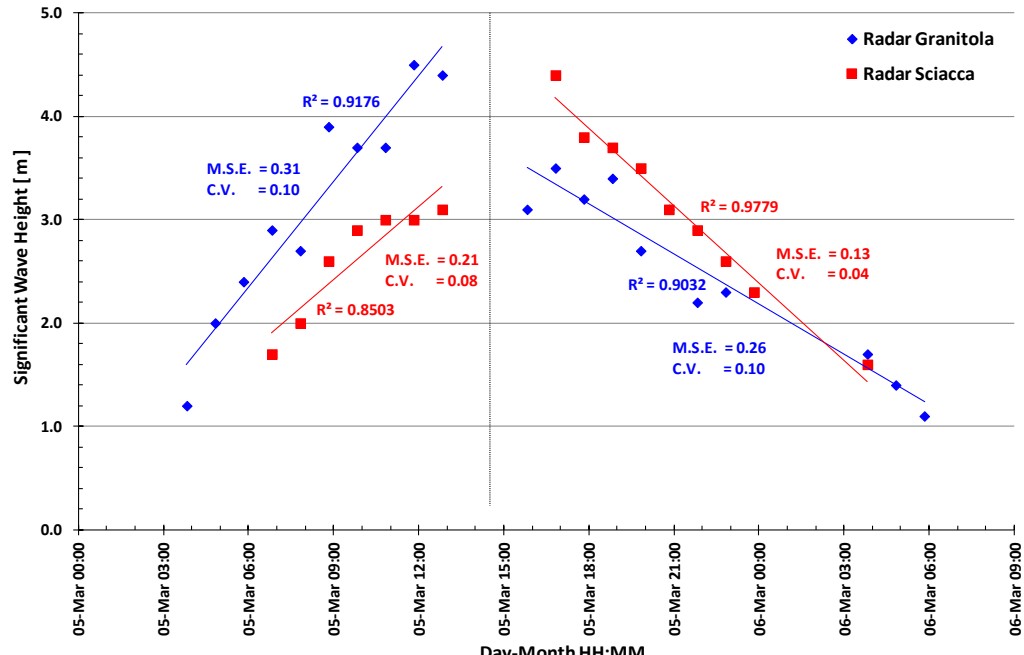

**Figure 11.** Best fit and scatter of radar data in Cape Granitola and Sciacca.

**Table 1.** Parameters of acquisition.

|  | Cape Granitola Site | Sciacca Site |
|---|---|---|
| **Antenna rotation period (Δt)** | 2.397 s | 2.097 s |
| **Radar coverage** | 1.24 Nm | 1.20 Nm |
| **Pixel Spacing (Δpsx, Δpsy)** | 4.48 m | 4.34 m |
| **Field of view** | ≈180° | ≈180° |
| **Antenna height (over sea level)** | 15 m | 25 m |
| **Number of image per dataset** | 32 | 32 |

**Table 2.** Variance of measurements for Event I.

|  | MSE (m) | CV |
|---|---|---|
| Buoy | 0.15 | 0.06 |
| Radar | 0.20 | 0.07 |



**Table 3.** Variance of measurements for Event II.

|  | **MSE (m)** | **CV** |
|---|---|---|
| Sciacca Ascending | 0.21 | 0.08 |
| Sciacca Descending | 0.13 | 0.04 |
| Cape Granitola Ascending | 0.31 | 0.10 |
| Cape Granitola Descending | 0.26 | 0.10 |