# Peer review of "Integration between X-Band Radar and Buoy Sea State Monitoring"

_Ocean Science, 2016_

## Referee Comment (RC1) · Anonymous Referee #1 · 7 Sep 2016

Review of "Integration between X-Band Radar and Buoy Sea State Monitoring"

by Ludeno et.al

General Comment:

In the manuscript, the authors reported the wave observation results obtained from X-band radar, buoy and a model. The technology has been mature for a long time. Thus, the originality of the presented work is little. In addition, the writing is poor.

Technical comments:

The title indicates integration of wave information from radar and buoy. However, it only contains simple comparison of all the sources obtained.

Another claimed contribution the manuscript is the confirmation of the consistency and

repeatability of the two X-band radars employed. This didn't provide any new technical contribution to the readers but confidence of the radar products.

On the second page, it is said "it is somewhat surprising that so few experiments are reported in the literature of combined application of two or more systems." This is not true, there many publications presenting the results of two or more systems.

Are the two calibration scale factors same for the two radar systems?

How the buoy wave data is linearly transformed should be explained.

The scatter plots of wave heights over time are not meaningful.

Editorial comments:

The manuscript is very poor-written. There are too many grammar mistakes and broken sentences. E.g.,

"begin the wave number" on page 3; "an U=(Ux, Uy) dis" on page 3; "An equalization step is implemented using the spectral Modulation Transfer Function (MTF), is to move from the filtered radar image spectrum ... by minimizing the electromagnetic modulation effect." On page 4 is broken...

---

## Referee Comment (RC2) · Anonymous Referee #2 · 22 Sep 2016

The authors did simultaneous measurements of waves during storm events with land-based X-band radars and wave buoy. Final target of this work seems to give some insight to Small Scale Storm Variation (SSSV) from the results of the measurements.

Radar and buoy measurements described in this work are both established methods (or tools), and there is no novelty and merit from this part.

The description of the two events is not shown in details. Are they induced by a similar weather system, or by a quite different situation? This must be an important information to discuss SSSV.

The reviewer was expected to learn about SSSV, but the description on this is very few. The authors should explain processes in a SSSV, and whether they are detectable from their deployment: two radars and a buoy. The authors claim that Eq. (5) can be used

as an index to discuss SSSV, but the background of this idea is not shown.

One of the advantages of radar measurement is to collect spatial distribution of backscatter from the wave field. The authors should try to assess spatial variability from their data to discuss SSSV.

(Overall) The authors should describe more on SSSV:

- what are the variations in a small scale storm

- are the variations detectable from the deployment: two radars and a buoy

- how to detect variations from the measured data

---

## Author Comment (AC1) · 22 Nov 2016

Review of "Integration between X-Band Radar and Buoy Sea State Monitoring"

by Ludeno et.al

General Comment:

In the manuscript, the authors reported the wave observation results obtained from X band radar, buoy and a model. The technology has been mature for a long time. Thus, the originality of the presented work is little. In addition, the writing is poor.

Technical comments:

The title indicates integration of wave information from radar and buoy. However, it only contains simple comparison of all the sources obtained.

Another claimed contribution the manuscript is the confirmation of the consistency and repeatability of the two X-band radars employed. This didn't provide any new technical contribution to the readers but confidence of the radar products.

Reply

We carried out the comparison between the two wave radar system and wave buoy at Capo Granitola and Sciacca sites, which are located in the south west part of Sicily . This area of the Mediterranean Sea has a significant biodiversity and is affected by several complex oceanographic processes. Therefore, the information about the sea state parameters as well as surface currents is important: to safeguard the biodiversity; to forecast the coastal erosion; to support decisions for the crisis events related to pollution. Although, the wave radar system and buoy are indeed established methods, we believe that the use of the devices simultaneously can be useful to understand these complex oceanographic processes.

In order to clarify this, the following sentences have been add at the end of the Introduction:

*"This area of the Mediterranean Sea has a significant biodiversity and is affected by several complex oceanographic processes. Therefore, the information about the sea state parameters as well as surface currents is important in order to safeguard the biodiversity, to forecast the coastal erosion, to support decisions for the crisis events related to pollution. The novelty of the work lies also in the possibility monitor these complex oceanographic processes simultaneously and to verify their evolution in the space and time on a small scale with various devices."*

On the second page, it is said "it is somewhat surprising that so few experiments are reported in the literature of combined application of two or more systems." This is not true, there many publications presenting the results of two or more systems.

Reply

To avoid misunderstanding we deleted this sentence.

Are the two calibration scale factors same for the two radar systems?

Reply

In Section 4 (Location and Data) of the submitted manuscript two slightly different wave radar systems are described. In particular, the Cape Granitola system is based on a CONSILIUM X Band radar, while a SPERRY Marine X-Band is installed at the Sciacca site. Consequently, as reported in page 6 line 4 of the submitted manuscript, the calibration scale factor depends on particular wave radar installation.

In order to clarify this, the following sentence has been added in the revised manuscript:

*"Hence, the wave spectra retrieved from the wave radar systems installed at Cape Granitola and Sciacca site have been calibrated with two different scale factor."*

How the buoy wave data is linearly transformed should be explained.

Reply

In order to clarify this point, a detailed explanation of the procedure has been added in the Section 5 :

*"If $H_1$, $H_2$, are the Wave Heights with a period $T_0$, their heights $H_1$ and $H_2$ at the depth $d_1$ and $d_2$ respectively, are linked by*

$$\frac{H_2{}^2}{H_1{}^2} = \frac{C_{g2}}{C_{g1}} = \frac{C_2}{C_1}\frac{n_2}{n_1} \qquad (5)$$

*Where C1, C2 are the wave velocities, given by the dispersion equation (2) reported above Cg1 and Cg2 are the group velocities and the parameter n is*

$$n = \frac{1}{2}\left(1 + \frac{2k}{sinh(2kd)}\right) \qquad (6)$$

*The dispersion equation cannot be solved directly, but well known approximation such as Hunt's (see for instance USACE Coastal Engineering Manual) are easily available"*

The scatter plots of wave heights over time are not meaningful.

Reply

The purpose of the plots of wave heights over time it is to show the dispersion of the wave height data around the trend. We agree with the referee that it is not a correlation. Therefore, in the revised manuscript, we have removed the correlation coefficient values and the term scatter plot has been changed in *"time regression"*.

Editorial comments:

The manuscript is very poor-written. There are too many grammar mistakes and broken sentences. E.g.,

"begin the wave number" on page 3; "an U=(Ux, Uy) dis" on page 3; "An equalization

step is implemented using the spectral Modulation Transfer Function (MTF), is to move

from the filtered radar image spectrum … by minimizing the electromagnetic modulation effect." On page 4 is broken:

Reply

We re-wrote part of the manuscript and corrected many grammar and style mistakes (in blue) and took into account many of the referee's suggestion (in green)

---

## Author Comment (AC2) · 22 Nov 2016

The authors did simultaneous measurements of waves during storm events with landbased X-band radars and wave buoy. Final target of this work seems to give some insight to Small Scale Storm Variation (SSSV) from the results of the measurements.

Radar and buoy measurements described in this work are both established methods (and tools), and there is no novelty and merit from this part.

Reply

The wave radar system and buoy are indeed established methods, but we believe that having applied two devices simultaneously to the same phenomenon has added something to the knowledge.

In order to clarify this, the following sentences have been add at the end of the Introduction:

"This area of the Mediterranean Sea has a significant biodiversity and is affected by several complex oceanographic processes. Therefore, the information about the sea state parameters as well as surface currents is important in order to safeguard the biodiversity, to forecast the coastal erosion, to support decisions for the crisis events related to pollution. The novelty of the work lies also in the possibility monitor these complex oceanographic processes simultaneously and to verify their evolution in the space and time on a small scale with various devices."

The description of the two events is not shown in details. Are they induced by a similar weather system, or by a quite different situation? This must be an important information to discuss SSSV.

Reply

The two storms are both caused by weather perturbations typical of the Mediterranean Sea. In the revised manuscript, we have added the evolution of the wind during the events from ECMWF data.

Figure 3a and Figure 7a in the revised manuscript

The reviewer was expected to learn about SSSV, but the description on this is very few. The authors should explain processes in a SSSV, and whether they are detectable from their deployment: two radars and a buoy. The authors claim that Eq. (5) can be used as an index to discuss SSSV, but the background of this idea is not shown.

(Overall) The authors should describe more on SSSV:

- what are the variations in a small scale storm

- are the variations detectable from the deployment: two radars and a buoy

- how to detect variations from the measured data

Reply

In the revised manuscript a brief discussion has been added with some references.

*"This dispersion (CV) is partly due to measurement errors, but it also provides some evidence of the presence of random oscillations of the storm intensity. This is an indicator of short term storm intensity variations (SSSV) also i.e. variation of wind intensity ("gustiness") on a much smaller scale than the evolution of the storm, and therefore inherently random in nature. Abdalla and Cavaleri (2002) actually simulated SSSV by feeding synthetic gusty wind series to a WAM model and provided some evidence of SSSV not unlike those found in the present work; later, Cavaleri and Burgers (1992) used the gustiness concept to improve the friction velocity estimates; Accadia et al (2007) made use of the concept in the context of Scatterometer wind observations while Pleskachevsky et al (2012) considered the influences of gustiness on ocean waves - in space rather than in time. Carratelli et al, 2014 gained some insight on the spatial aspect phenomenon by analysing the along-track altimeter values. So far little or no experimental work has been carried out on the time behaviour of wave data from either buoy or x-band radar*

*The similar values for MSE and CV of the very same storm with two entirely different measuring systems opens new possibilities for the analysis of SSSV.*

*Unfortunately, very little can be said about the remaining part of the storm, since the wave meter buoy was not operating at the time; the only useful information are the MSE, M and CV values, as above which are shown in Figure 6, and appear to be consistent with the previous ones."*

One of the advantages of radar measurement is to collect spatial distribution of backscatter from the wave field. The authors should try to assess spatial variability from their data to discuss SSSV.

Reply

In this work, we did not use the radar to derive the spatial distribution but only temporal variation. The area covered by the radar is however (2 km$^2$) too small to provide any understanding of gustiness